# Predictive value of heart rate variability on long-term mortality in end-stage kidney disease on hemodialysis

Nichanan Osataphan[1], Wanwarang Wongcharoen[1], Arintaya Phrommintikul[1], Phasakorn Putchagarn[2], Kajohnsak Noppakun[3,4]*

1 Division of Cardiology, Department of Internal Medicine, Faculty of Medicine, Chiang Mai University, Chiang Mai, Thailand, 2 Uttaradit Hospital, Uttaradit, Thailand, 3 Division of Nephrology, Department of Internal Medicine, Faculty of Medicine, Chiang Mai University, Chiang Mai, Thailand, 4 Pharmacoepidemiology and Statistics Research Center (PESRC), Faculty of Pharmacy, Chiang Mai University, Chiang Mai, Thailand

* kajohnsak.noppakun@cmu.ac.th

**Data Availability Statement:** The informed consent given by the participants does not cover data posted in public databases. However, data available upon request should be sent to

## Abstract

Autonomic disturbance is common in end-stage kidney disease (ESKD). Heart rate variability (HRV) is a useful tool to assess autonomic function. We aimed to evaluate the predictive value of HRV on all-cause mortality and explore the proper timing of HRV assessment. This prospective cohort study enrolled 163 ESKD on hemodialysis patients from April-December 2018. HRV measurements were recorded ten minutes before hemodialysis, four hours during hemodialysis, and ten minutes after hemodialysis. Clinical parameters and all-cause mortality were recorded. Cox-proportional hazard regression was used for statistical analysis. After a median follow up of 40 months, 37 (22.7%) patients died. Post-dialysis HRV parameters including higher very low frequency (VLF) (hazard ratio [HR], 0.881; 95%confidence interval [CI], 0.828–0.937; p<0.001), higher normalized low frequency (nLF) (HR, 0.950; 95%CI, 0.917–0.984; p = 0.005) and higher LF/HF ratio (HR, 0.232; 95%CI, 0.087–0.619; p = 0.004) were the independent predictors associated with lower risk for all-cause mortality. Higher post-dialysis normalized high frequency (nHF) increased risk of mortality (HR, 1.051; 95%CI, 1.015–1.089; p = 0.005). HRV parameters at pre-dialysis and during dialysis were not predictive for all-cause mortality. The area under receiver operating characteristic curve (AuROC) of VLF for survival was highest compared to other HRV parameters at post-dialysis period (AuROC 0.71; 95% CI; 0.62–0.79; p<0.001). In conclusion, post-dialysis HRV parameters predicted all-cause mortaliy in ESKD. VLF measured at post-dialysis exhibited best predictive value for survival in chronic hemodialysis patients.

## Introduction

Cardiovascular disease is the major cause of death in patients with end-stage kidney disease (ESKD). Autonomic disturbance has been proposed as one of the main mechanisms of sudden cardiac death in ESKD. The imbalance of the autonomic system is common and can be found

kajohnsak.noppakun@cmu.ac.th or the ethics committee of Faculty of Medicine, Chiang Mai University (researchmed@cmu.ac.th).

**Funding:** This work was supported by the Faculty of Medicine Endowment Fund for Medical Research, Chiang Mai University, Thailand (141/2561). The funder had no role in study design, data collection, and analysis, decision to publish, or preparation of the manuscript.

**Competing interests:** All the authors declare that they have no competing interests.

in up to 50% of ESKD [1]. The possible mechanisms of autonomic dysfunction in chronic kidney disease are associated with impaired reflex control of autonomic activity, activation of the renin-angiotensin-aldosterone system, and structural remodeling of the heart [2]. These effects lead to an overactivation of the sympathetic system and a reduction of parasympathetic activity [2]. Increased sympathetic activation contributes to a rise in basal heart rate, and promotes cardiac remodeling and fibrosis which may also have negative effects on cardiac function [3].

Measuring heart rate variability (HRV) is a non-invasive method that indirectly evaluates autonomic function on the cardiac sinus node. HRV represents the oscillation between consecutive heartbeats. In a normal heart, HRV fluctuates with respiration which reflects the interaction between the sympathetic and parasympathetic systems. HRV has been shown to predict mortality in many diseases, most commonly following myocardial infarction [4]. Consistently, HRV has also been used to predict cardiovascular and all-cause mortality in patients with ESKD on hemodialysis [5]. However, the data on the predictive values of HRV parameters on mortality in the hemodialysis patients remained inconsistent [6–8]. Several studies showed that a decrease in the ratio of low frequency (LF) to high frequency (HF) was associated with increased all-cause mortality [7, 8]. In contrast, a study by Kuo *et al.* showed that a higher LF to HF ratio was an independent risk factor of mortality [6]. The diversity of selected HRV parameters and the different timing for HRV assessment may confer inconsistent findings in the hemodialysis population [9, 10]. Additionally, the appropriate time point for HRV measurement was not well established. There were only two previous studies that performed repeated HRV measurement on the dialysis day. Chen *et al.* measured HRV before and 30 minutes after hemodialysis session. They found that the change of LF between the pre-hemodialysis and post-hemodialysis predicted overall and cardiovascular mortality [10]. While a more recent study by Chang *et al.* performed the repeated HRV measurements before and during hemodialysis and found that HRV parameters during hemodialysis were predictive for cardiovascular mortality [9]. To the best of our knowledge, there was no previous study that performed repeated HRV measurement at 3-time points (pre-dialysis, during dialysis and post-dialysis). As a result, the aims of our study were to determine the best timing for HRV measurement by performing repeated HRV analysis at 3-time points and to determine the best predictive HRV parameter on all-cause mortality.

## Materials and methods

### Study design and participants

This was a prospective study using a cohort of 163 ESKD patients on hemodialysis previously enrolled at Chiang Mai University Hospital between April 2018 to December 2018. The inclusion criteria were age more than 18 years old and on maintenance hemodialysis three times per week. Patients with atrial fibrillation, atrial flutter, and implanted pacemaker were excluded due to the limitation of HRV assessment. Informed consent was obtained from all participants. Blood pressure was measured before hemodialysis in a seated position after resting for more than 5 minutes using validated automatic oscillometric device (OMRON HEM-7121 automatic blood pressure monitor). Blood samples for complete blood count, creatinine, electrolytes and albumin were collected before the initiation of hemodialysis. Blood pressure was repeated after completion of hemodialysis session in the same manner. Blood for creatinine and electrolytes were collected again after dialysis as part of hemodialysis care protocol. Before starting hemodialysis, a bedside echocardiogram was performed. The modified biplane Simpson's technique was used to calculate the left ventricular ejection fraction (LVEF). Holter monitoring (GE Seer Light Extend, GE Medical Systems, Suzuken Company, Ltd.) was used for HRV analysis. A large observational study indicated that all-cause and cardiovascular

mortality were higher during long interdialytic days [11]. Accordingly, we selected the HRV parameters during a long interdialytic interval for analysis (three-day interval between hemodialysis sessions). Holter recording lasted ten minutes before hemodialysis and continued during the four-hour hemodialysis session and ten minutes after hemodialysis. The baseline characteristics including demographic data, co-morbidities, medications, and laboratory results were collected. The study was carried out following the Declaration of Helsinki and was approved by the Research Ethics Committee of the Faculty of Medicine, Chiang Mai University, Study Code MED-2564-08420. This study was registered in the Thai Clinical Trials Registry (TCTR), identification number TCTR20220215008.

## HRV analysis

The electrocardiogram (ECG) was manually preprocessed before data analysis. The premature supraventricular beats, ventricular beats, pauses and missed beats were filtered and replaced by interpolated values. Frequency-domain analyses were performed according to standard guidelines [12]. Frequency-domain HRV was analyzed using autoregressive power spectral analysis applied to the RR interval time series. The following spectral bands were identified: very low frequency (VLF) (0.003–0.04 Hz), low frequency (LF) (0.04–0.15 Hz), and high frequency (HF) (0.15–0.4 Hz). Total power (0–0.5 Hz) and the areas below each peak were calculated in absolute units ($ms^2$). The normalized LF (nLF) and normalized HF (nHF) were calculated as a percentage. The nLF was the proportion of LF to the LF plus HF (nLF = LF*100/(LF + HF)) and nHF was the proportion of HF to the LF plus HF (nHF = HF*100/(LF + HF)).

## Outcomes

The primary endpoint of this study was the predictive value of HRV parameters at different time points on all-cause mortality. All patients were followed until death and the remainder were followed until September 2021. The survivors were censored at the end of the follow-up. Patients who were lost to follow-up were censored at the last visit date. Patients who underwent kidney transplantation were censored at the transplant date.

## Statistical analysis

The categorical data were presented as N (%) and compared between groups with Fisher's exact test. The continuous data were presented as mean ± SD or median (interquartile range) and compared between groups with Student's t-test or Mann-Whitney U test where appropriate. Survival analysis was performed using Cox proportional hazards regression. The predictive value of HRV parameters on all-cause mortality was determined as hazard ratio (HR) and 95% confidence interval (CI). Univariable clinical risk factors with a p-value < 0.10 were included in multivariable Cox proportional hazard regression with the HRV parameters. Kaplan-Meier survival curve of the selected parameters was compared using the log-rank test. Area under receiver operating characteristic curve (AuROC) was performed to determine predictive value of HRV parameters. Each ROC was compared using the algorithm suggested by DeLong and Clarke-Pearson [13]. The p-value of <0.05 was considered statistically significant. All statistical analyses were performed with Stata software version 16.

## Results

### Baseline characteristics

One hundred and sixty-three patients with ESKD on hemodialysis enrolled in the study. After a median follow-up of 40 months (IQR 35.1–40.8), 37 (22.7%) died. Of the 37 patients who

died, 7 were from cardiovascular causes, 9 were from infection, 3 were from metabolic causes, 13 had an unknown cause and 5 had other causes. For the cardiovascular causes, 3 had heart failure, 3 had sudden cardiac death and 1 had acute myocardial infarction. Baseline characteristics between survivors and non-survivors groups are shown in Table 1. The majority of

**Table 1. Baseline characteristics between survivors and non-survivors.**

| | Survivors (n = 126) | Non-survivors (n = 37) | P-value |
|---|---|---|---|
| Age (year) | 58.74 ± 11.94 | 70.32 ± 13.92 | <0.001 |
| Male (%) | 69 (54.76%) | 19 (51.35%) | 0.85 |
| LVEF (%) | 64.15 ± 11.51 | 58.84 ± 14.0 | 0.023 |
| Causes of ESKD | | | 0.78 |
| • Diabetic nephropathy | 55 (43.65%) | 17 (45.95%) | |
| • Hypertension | 25 (19.84%) | 7 (18.92%) | |
| • Glomerulonephritis | 20 (15.87%) | 3 (8.11%) | |
| • Obstructive uropathy | 8 (6.35%) | 3 (8.11%) | |
| • Others | 18 (14.29%) | 7 (18.92%) | |
| **Comorbidities** | | | |
| Hypertension (%) | 117 (94.35%) | 34 (94.44%) | 1.00 |
| Diabetes mellitus (%) | 63 (50.00%) | 19 (51.35%) | 1.00 |
| Dyslipidemia (%) | 85 (69.67%) | 26 (72.22%) | 0.838 |
| Coronary artery disease (%) | 14 (11.48%) | 10 (27.78%) | 0.031 |
| Peripheral artery disease (%) | 2 (1.64%) | 2 (5.56%) | 0.223 |
| Chronic obstructive pulmonary disease (%) | 2 (1.64%) | 1 (2.78%) | 0.542 |
| **Medications** | | | |
| ACEI/ARB (%) | 39 (31.97%) | 17 (48.57%) | 0.076 |
| Beta-blockers (%) | 75 (61.48%) | 29 (82.86%) | 0.025 |
| Calcium channel blockers (%) | 90 (73.77%) | 27 (77.14%) | 0.827 |
| Diuretics (%) | 57 (46.72%) | 18 (51.43%) | 0.702 |
| Alpha blockers (%) | 41 (33.61%) | 11 (31.43%) | 1.00 |
| Statins (%) | 84 (68.85%) | 23 (65.71%) | 0.837 |
| Anti-platelets (%) | 44 (36.07%) | 16 (45.71%) | 0.328 |
| Oral anticoagulants (%) | 6 (4.92%) | 5 (14.29%) | 0.069 |
| **Hemodialysis data** | | | |
| Dialysis vintage (median, IQR) | 3 (1–5) | 4 (2–6) | 0.17 |
| Kt/V | 1.66 ± 0.34 | 1.66 ± 0.32 | 0.99 |
| Fluid removal per session (ml) | 2449.57 ± 93.53 | 2284 ± 145.90 | 0.38 |
| **Clinical and biochemical data** | | | |
| Pre-dialysis SBP (mmHg) | 146.90 ± 19.87 | 141.70 ± 18.89 | 0.16 |
| Pre-dialysis DBP (mmHg) | 76.76 ± 12.99 | 66.94 ± 11.92 | <0.001 |
| Post-dialysis SBP (mmHg) | 145.30 ± 19.43 | 143.46 ± 13.87 | 0.597 |
| Post-dialysis DBP (mmHg) | 78.29 ± 12.16 | 72.05 ± 10.30 | 0.005 |
| Interdialytic weight gain (kg) | 2.16 ± 1.05 | 2.09 ± 0.81 | 0.73 |
| Net ultrafiltration (mL) | 2449.51 ± 1049.92 | 2284 ± 887.5 | 0.38 |
| Albumin (g/dL) | 3.98 ± 0.45 | 3.77 ± 0.40 | 0.018 |
| Hemoglobin (g/dL) | 10.60 ± 1.59 | 10.03 ± 2.00 | 0.07 |
| Pre-dialysis serum sodium (mmol/L) | 137.12 ± 3.43 | 136.11 ± 2.93 | 0.105 |
| Pre-dialysis serum potassium (mmol/L) | 4.36 ± 0.62 | 4.21 ± 0.60 | 0.17 |

Data were presented as mean ± SD.

ACEI/ARB, angiotensin-converting enzyme inhibitor/angiotensin receptor blocker; DBP, diastolic blood pressure; ESKD, end-stage kidney disease; LVEF, left ventricular ejection fraction; SBP, systolic blood pressure.

patients were male. Non-survivors were significantly older than the survivors (70.32 ± 13.92 years vs 58.74 ± 11.94 years, p < 0.001). Mean left ventricular ejection fraction (LVEF) was significantly lower in the non-survivors compared to the survivors (58.83 ± 14.0% vs 64.15 ± 11.51%, p = 0.023). Almost all patients had hypertension and about half of them had diabetes mellitus. The prevalence of coronary artery disease was higher in the non-survivors compared to the survivors (27.78% vs 11.48%, p = 0.031). The use of ACEI/ARB was not significantly different between groups (48.57% vs 31.97%, p = 0.076) while the use of beta-blockers was significantly higher in the non-survivors than in the survivors (82.86% vs 61.48%, p = 0.025). Pre-dialysis systolic blood pressure (SBP) was similar between the non-survivors and the survivors (141.70 ± 18.89 mmHg vs 146.90 ± 19.87 mmHg, p = 0.16). On the contrary, the pre-dialysis diastolic blood pressure (DBP) was lower in the non-survivors compared to the survivors (66.94 ± 11.92 mmHg vs 76.76 ± 12.99 mmHg, p < 0.001). The post-dialysis DBP was also lower in the non-survivors than the survivors (72.05 ± 10.30 mmHg vs 78.29 ± 12.16 mmHg, p = 0.005). Non-survivors had lower serum albumin than the survivors (3.77 ± 0.40 g/dL vs 3.98 ± 0.45 g/dL, p = 0.018). Other biochemical data including interdialytic weight gain, net ultrafiltration, and pre-dialysis serum sodium and potassium were similar between the two groups.

## Predictive values of HRV parameters on long-term mortality

The comparison of HRV parameters between the survivors and the non-survivors is presented in Table 2. We analyzed the HRV parameters at three time points including pre-dialysis, during dialysis, and post-dialysis. During the pre-dialysis period, VLF was significantly lower in the non-survivors compared to the survivors ($10.46 ± 5.70$ ms$^2$ vs $13.31 ± 7.76$ ms$^2$, p = 0.039). During the dialysis period, the non-survivors had significantly lower nLF value compared to the survivors (49.44 ± 10.95 vs 53.43 ± 10.61, p = 0.047). In opposition to the nLF, the nHF value during dialysis was higher in the non-survivors than in the survivors (50.56 ± 10.95 vs 46.56 ± 10.61, p = 0.047). Interestingly, during the post-dialysis period, all HRV values were significantly different between the two groups. The VLF, nLF, and LF/HF were lower in the

**Table 2. Comparison of heart rate variability (HRV) parameters between survivors and non-survivors.**

| HRV parameters | Survivors (N = 126) | Non-survivors (N = 37) | P-value |
|---|---|---|---|
| **Pre-dialysis** | | | |
| VLF (ms$^2$) | 13.31 ± 7.76 | 10.46 ± 5.70 | 0.039 |
| nLF | 51.66 ± 11.70 | 48.58 ± 9.50 | 0.14 |
| nHF | 48.34 ± 11.70 | 51.42 ± 9.50 | 0.14 |
| LF/HF ratio | 1.20 ± 0.56 | 1.01 ± 0.40 | 0.06 |
| **During dialysis** | | | |
| VLF (ms$^2$) | 16.26 ± 8.97 | 13.25 ± 8.71 | 0.07 |
| nLF | 53.43 ± 10.61 | 49.44 ± 10.95 | 0.047 |
| nHF | 46.56 ± 10.61 | 50.56 ± 10.95 | 0.047 |
| LF/HF ratio | 1.24 ± 0.49 | 1.07 ± 0.47 | 0.07 |
| **Post-dialysis** | | | |
| VLF (ms$^2$) | 16.46 ± 10.69 | 9.83 ± 5.33 | <0.001 |
| nLF | 53.00 ± 11.13 | 47.73 ± 11.62 | 0.013 |
| nHF | 47.00 ± 11.13 | 52.27 ± 11.62 | 0.013 |
| LF/HF ratio | 1.25 ± 0.55 | 0.99 ± 0.49 | 0.010 |

LF/HF ratio, low frequency/high-frequency ratio; nHF, normalized high frequency; nLF, normalized low frequency; VLF, very low frequency.

**Table 3. Univariable and multivariable analysis of heart rate variability (HRV) parameter and all-cause mortality.**

| HRV parameters | Univariable | | Multivariable[*] | |
|---|---|---|---|---|
| | HR (95% CI) | P-value | HR (95% CI) | P-value |
| **Pre-dialysis** | | | | |
| VLF (ms$^2$) | 0.949 (0.901–1.000) | 0.052 | 0.949 (0.898–1.002) | 0.063 |
| nLF | 0.981 (0.954–1.008) | 0.185 | 0.988 (0.952–1.025) | 0.536 |
| nHF | 1.018 (0.990–1.047) | 0.185 | 1.011 (0.975–1.049) | 0.536 |
| LF/HF ratio | 0.551 (0.282–1.076) | 0.081 | 0.717 (0.299–1.719) | 0.456 |
| **During dialysis** | | | | |
| VLF (ms$^2$) | 0.963 (0.923–1.003) | 0.081 | 0.962 (0.921–1.005) | 0.084 |
| nLF | 0.972 (0.945–1.000) | 0.055 | 0.966 (0.931–1.003) | 0.078 |
| nHF | 1.028 (0.999–1.057) | 0.055 | 1.034 (0.996–1.073) | 0.078 |
| LF/HF ratio | 0.530 (0.252–1.072) | 0.084 | 0.476 (0.183–1.239) | 0.128 |
| **Post-dialysis** | | | | |
| VLF (ms$^2$) | 0.912 (0.866–0.960) | 0.001 | 0.881 (0.828–0.937) | <0.001 |
| nLF | 0.968 (0.943–0.995) | 0.021 | 0.950 (0.917–0.984) | 0.005 |
| nHF | 1.032 (1.004–1.060) | 0.021 | 1.051 (1.015–1.089) | 0.005 |
| LF/HF ratio | 0.419 (0.208–0.843) | 0.015 | 0.232 (0.087–0.619) | 0.004 |

[*]Adjusted for age, LVEF, albumin, hemoglobin, coronary artery disease, use of beta-blockers, use of ACEI/ARB, use of oral anticoagulation, and pre-dialysis DBP.
ACEI/ARB, angiotensin-converting enzyme inhibitor/angiotensin receptor blocker; DBP, diastolic blood pressure; LF/HF ratio, low frequency/high-frequency ratio; LVEF, left ventricular ejection fraction; nHF, normalized high frequency; nLF, normalized low frequency; VLF, very low frequency.

non-survivors compared to the survivors (p < 0.001, p = 0.013, and p = 0.01, respectively). The nHF was the only parameter that had a significantly higher value in the non-survivors than the survivors during the post-dialysis period (52.27 ± 11.62 vs 47.00 ± 11.13, p = 0.013).

The univariable and multivariable Cox regression analysis of HRV parameters on all-cause mortality is shown in Table 3. During pre-dialysis and the dialysis period, none of the HRV parameters were predictive of all-cause mortality. During the post-dialysis period, all HRV parameters were predictive of all-cause mortality and remained statistically significant after adjusting for clinical factors. In multivariable analysis, the higher VLF and nLF values were associated with lower mortality (HR, 0.881; 95% CI, 0.828–0.937; p < 0.001 and HR, 0.950; 95% CI, 0.917–0.984; p = 0.005). In conjunction with VLF and nLF, the increased LF/HF ratio was associated with a lower risk of all-cause mortality (HR, 0.232; 95% CI, 0.087–0.619; p = 0.004). On the other hand, the increased nHF value at post-dialysis was predictive of all-cause mortality (HR, 1.051; 95%CI, 1.015–1.089; p = 0.005). Finally, we performed the Kaplan-Meier survival curve analysis according to the post-dialysis HRV parameters (Fig 1). The post-dilaysis VLF, nLF and LF/HF ratio lower than the median value was predictive for mortality (HR, 3.50; 95%CI, 1.65–7.41; p<0.001, HR 2.25; 95%CI, 1.13–4.47; p = 0.02 and HR 2.55; 95% CI, 1.26–5.18; p = 0.006, respectively). nHF at median value or higher was also predictive for mortality (HR 2.25; 95%CI, 1.13–4.47; p = 0.02).

## AuROC of HRV parameters

We compared the predictive value of HRV parameters by performing AuROC. The parameters were directly compared at the same period (pre-dialysis, during dialysis or post-dialysis). Each parameter was also compared at 3-time points. The results were presented in Table 4. There was no significant difference in the AuROC before and during dialysis. Interestingly, there was a significant difference in the AuROC at post-dialysis period. The AuROC of VLF

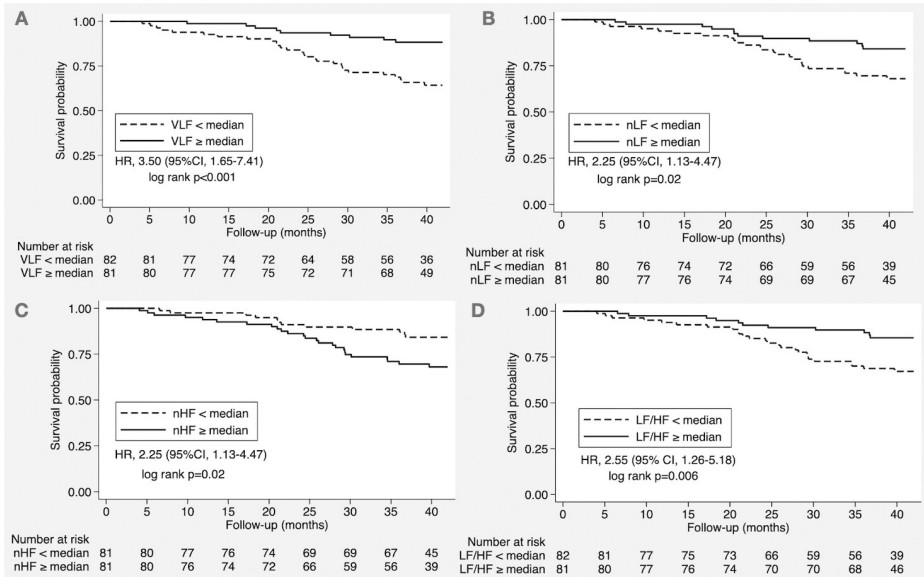

**Fig 1. All-cause survival curve according to the post-dialysis HRV parameters.** Panel A: VLF, Panel B: nLF, Panel C: nHF and Panel D: LF/HF ratio.

for survival was highest compared to other HRV parameters at post-dialysis period (AuROC 0.71; 95% CI; 0.62–0.79; p<0.001). Post-dialysis VLF had highest AuROC for survival compared to pre-dialysis VLF and VLF during dialysis (p = 0.008).

## Discussion

This study evaluated the predictive value of HRV parameters on all-cause mortality at different time points including pre-dialysis, during dialysis, and post-dialysis. After a median follow-up of 40.3 months, death occurred in 37 patients (22.7%). We identified HRV parameters including decreased VLF, decreased nLF, increased nHF, and decreased LF/HF ratio in the post-dialysis period as the independent predictors of higher all-cause mortality. Nevertheless, the measurement of these parameters at pre-dialysis and during the dialysis period were not

**Table 4. Area under receiver operating characteristic curve (AuROC) for survival according to VLF, nLF and LF/HF ratio and for death according to nHF at different time points.**

| HRV parameters | Pre-dialysis | During Dialysis | Post-dialysis | p-value* |
|---|---|---|---|---|
| | AuROC | AuROC | AuROC | |
| | (95%CI) | (95%CI) | (95%CI) | |
| VLF (ms$^2$) | 0.60 (0.50–0.70) | 0.61 (0.51–0.72) | 0.71 (0.62–0.79) | 0.008 |
| nLF | 0.58 (0.49–0.68) | 0.60 (0.50–0.71) | 0.64 (0.53–0.74) | 0.50 |
| nHF | 0.58 (0.49–0.68) | 0.60 (0.50–0.71) | 0.64 (0.53–0.74) | 0.50 |
| LF/HF ratio | 0.59 (0.49–0.69) | 0.60 (0.49–0.70) | 0.65 (0.55–0.76) | 0.24 |
| **p-value**** | 0.13 | 0.11 | <0.001 | |

*P-value comparing dialysis periods for each HRV parameter.

** P-value comparing HRV parameters for each dialysis period.

LF/HF ratio, low frequency/high-frequency ratio; nHF, normalized high frequency; nLF, normalized low frequency; VLF, very low frequency.

predictive of all-cause mortality. Further analysis by AuROC identified post-dialysis VLF as the best predictor for survival.

HRV is a useful tool to determine autonomic dysfunction which is one of the main mechanisms responsible for sudden cardiac death in ESKD. In our study, the HRV parameters at pre-dialysis and during dialysis were not predictive of mortality. The fluid shift during hemodialysis may affect the change of autonomic function more prominently in the post-dialysis period. As a result, the post-dialysis period is possibly the most appropriate timing of HRV measurement for the prognostic purpose in patients with chronic hemodialysis. There were only two previous studies that performed repeated HRV measurements on the dialysis day [9, 10]. The first study by *Chen* et al. calculated a difference in HRV parameters between pre-dialysis and post-dialysis, decreased in LF was shown to be predictive of overall and cardiovascular mortality [10]. Another study by *Chang* et al. performed the repeated HRV measurement at pre-dialysis and during dialysis but did not perform the post-dialysis measurement [9]. They identified the decreased VLF and LF/HF ratio during dialysis as the independent predictors for cardiovascular mortality. Our study was the first study that measured HRV parameters at 3-time points including pre-dialysis, during and post-dialysis. The post-dialysis VLF had best predictive ability for survival (AuROC = 0.71). Furthermore, post-dialysis VLF lower than the median value ($<12.99$ ms$^2$) had the 3.5-fold increased risk of mortality (Fig 1). This cutoff value can be used to identify the hemodialysis patients with heightened risk, to whom attentive care should be provided.

VLF is generated by several factors including the thermoregulatory and renin-angiotensin system and the physiologic mechanisms responsible for VLF activity are uncertain [14, 15]. The predictive value of VLF on mortality in ESKD was less studied compared to other frequency domain parameters [5]. A previous study showed that lower VLF was predictive of major cardiovascular events and hospitalization in chronic hemodialysis patients [16]. A more recent study by *Chang* et al. also found that the decreased VLF was an independent predictor of cardiovascular mortality in chronic hemodialysis patients [9]. The findings from these two previous studies were consistent with our result. Only a single study demonstrated a higher VLF as a risk factor for all-cause mortality in ESKD patients [17]. Future studies should explore the predictive value of VLF and a plausible mechanistic explanation of its association with mortality in chronic hemodialysis.

Concerning LF and HF, previous study showed that LF power reflexed a baroreceptor function [18]. The elevated HF power indicates an increased parasympathetic or decreased sympathetic activity and the LF/HF ratio reflects the sympathovagal balance [9, 14]. Generally, acute removal of fluid during hemodialysis activates the baroreflex system and increases sympathetic activity. Interestingly, we demonstrated that nLF was reduced and the nHF was increased at post-diaysis which consequently resulted in the decrease in LF/HF ratio in the non-survivors. These findings suggested that the non-survivors possibly had a baroreceptor failure and a blunted sympathetic response after fluid removal, which may have been related to the poor prognosis.

This study has several limitations. There is only a small number of patients with CV death. Therefore, we did not perform a statistical analysis of the predictive value of HRV parameters on cardiovascular mortality. This may not reflect the true effects of autonomic function on the cardiovascular outcome as it is proposed to be associated with sudden cardiac death and cardiovascular events. In addition, we had a relatively shorter follow-up period compared to a previous study [9]. However, we were still able to identify several independent parameters for mortality as mentioned. Some potential variables associated with cardiovascular mortality were not collected including the residual renal function and QT interval.

## Conclusions

Our study highlighted the beneficial role of HRV measurement as a non-invasive tool to predict long-term mortality in chronic hemodialysis patients. We identified that the lower VLF, lower nLF, higher nHF, and lower LF/HF ratio during the post-hemodialysis period were the independent predictors associated with higher all-cause mortality. Post-dialysis period could be the most appropriate timing for HRV measurement in this group of patients. The post-dialysis VLF had the best predictive value for survival in chronic hemodialysis patients.

## Acknowledgments

We would like to express our appreciation for all the efforts and contributions to the study support from Staff in the Northern Dialysis Center and Division of Nephrology, Department of Internal Medicine, Faculty of Medicine, Chiang Mai University, Chiang Mai, Thailand.

## Author Contributions

**Conceptualization:** Nichanan Osataphan, Wanwarang Wongcharoen, Kajohnsak Noppakun.

**Data curation:** Arintaya Phrommintikul, Phasakorn Putchagarn, Kajohnsak Noppakun.

**Formal analysis:** Nichanan Osataphan, Wanwarang Wongcharoen, Arintaya Phrommintikul, Kajohnsak Noppakun.

**Funding acquisition:** Wanwarang Wongcharoen.

**Investigation:** Phasakorn Putchagarn.

**Methodology:** Arintaya Phrommintikul, Phasakorn Putchagarn.

**Supervision:** Nichanan Osataphan, Wanwarang Wongcharoen.

**Validation:** Kajohnsak Noppakun.

**Writing – original draft:** Nichanan Osataphan, Kajohnsak Noppakun.

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
