## [Decision Letter · Decision Letter 0]

1 Dec 2022

PONE-D-22-29950Predictive Value of Heart Rate Variability on Long-Term Mortality in End-Stage Renal Disease on HemodialysisPLOS ONE

Dear Dr. Noppakun,

Thank you for submitting your manuscript to PLOS ONE. After careful consideration, we feel that it has merit but does not fully meet PLOS ONE’s publication criteria as it currently stands. Therefore, we invite you to submit a revised version of the manuscript that addresses the points raised during the review process.

We look forward to receiving your revised manuscript.

Kind regards,

Eyüp Serhat Çalık

Academic Editor

PLOS ONE

Journal Requirements:

4. Please remove your figures from within your manuscript file, leaving only the individual TIFF/EPS image files, uploaded separately. These will be automatically included in the reviewers’ PDF.

Additional Editor Comments :

Congratulations to the authors for this interesting study. Overall, we think it's a well-written manuscript. The manuscript was evaluated by three reviewers. Their comments on the manuscript are below. We think that your article will be of better quality in the light of those suggestions.

Reviewers' comments:

Reviewer's Responses to Questions

**Comments to the Author**

1. Is the manuscript technically sound, and do the data support the conclusions?

Reviewer #1: Partly

Reviewer #2: Partly

Reviewer #3: Partly

2. Has the statistical analysis been performed appropriately and rigorously? 

Reviewer #1: Yes

Reviewer #2: No

Reviewer #3: No

3. Have the authors made all data underlying the findings in their manuscript fully available?

Reviewer #1: Yes

Reviewer #2: Yes

Reviewer #3: No

4. Is the manuscript presented in an intelligible fashion and written in standard English?

Reviewer #1: Yes

Reviewer #2: Yes

Reviewer #3: No

5. Review Comments to the Author

Reviewer #1: In the study, the authors evaluated the predictive value of HRV on all-cause mortality and also explored the proper timing of HRV assessment in patients on dialysis.

It is well known that autonomic disturbance is common in such population. The findings are interesting. However, the study enrolled a very small number of patients.

In addition, important information are lacking in the present form.

Specific comments:

1. How did the authors check LVEF? If UCG was used, other important parameters such as LVMI, E/e’ LAD and so on might affect clinical outcomes in patients on dialysis.

2. QT duration is associated with incidence of critical arrhythmia.

3. Etiology of renal disease should be added.

4. How did the authors measure blood pressure and laboratory data? Please explain.

5. During follow-up phase, death occurred in 37 patients. Please add precise causes of deaths in such patients.

6. Please add definitions of comorbidity.

Reviewer #2: Thank you for the opportunity to review this study.

This study examines the predictive ability of heart rate variability before, during, and after dialysis for mortality, respectively.

It is a very interesting study, but I have a few concerns.

1.

After line 57, the authors introduce a previous study on the association between heart rate variability and mortality in hemodialysis patients. The authors state that the results are inconsistent, but it would be better to further organize the information from existing studies. Fig. 4 of the meta-analysis (reference #5) cited by the authors shows that studies with long-term measurements show less heterogeneity. The results of studies with short-term measurements similar to this study are inconsistent. If this study focuses its research question on which of the short-term measurements has better predictive ability, then it would be better to organize the information by focusing on timing rather than discussing parameter issues, so that the reader can clearly see what is known and what is not known. If the focus is on the question of whether a study has better predictive ability, then organizing the information to focus on timing rather than discussing parameter issues should clearly communicate to readers what is known and what is not known.

2.

The authors chose to employ a variable with a p-value <0.10 in the univariate analysis as the adjustment variable in the multivariate analysis; Table 1 shows that diastolic blood pressure was not employed for post-dialysis, despite the fact that p<0.10 for both pre-dialysis and post-dialysis. Why were both not entered into the model? If these show multicolinearity, show the data. Also, it seems odd that they used pre-dialysis diastolic blood pressure to adjust for post-dialysis heart rate variability when the final results show that post-dialysis HRV has better predictive power.

3．

The authors mention in the discussion that the amount of fluid removal by dialysis is related to heart rate volatility, but there are several studies that show that residual renal function and the amount of fluid removed are themselves related to mortality. Shouldn't these be included as adjustment variables?

4.

4.

The authors present the results of Table 3 and conclude that the post-dialysis parameters were predictive. The authors seem to have reached this conclusion because only the post-dialysis parameter showed a statistically significant difference, but it is questionable whether this is due to the size of the effect size or the sample size. Second, it is also questionable whether this conclusion can be reached without a direct statistical comparison of each HRV parameter. If the authors want to compare the predictive power of the parameters, they should obtain the c-index and compare each parameter to each other.

5.

There is a typo error in the reference 7.

Reviewer #3: This study evaluated the predictive value of heart rate variability (HRV) on all-cause mortality and explore the proper timing of HRV assessment. Clinical parameters and all-cause mortality were analyzed using cox proportional hazard regression. They found that post-dialysis HRV parameters including higher very low frequency (VLF), higher normalized low frequency (nLF) and higher LF/HF ratio were the independent predictors associated with lower risk for all-cause mortality, and that higher post-dialysis normalized high frequency (nHF) increased the risk of mortality. They concluded that HRV parameters predicted all-cause mortaliy in end-stage renal disease patients ESRD. This is an interesting study. However, some concerns need to be addressed.

Major concerns

1. Table 1. The age of non-survivors (70.32 ± 13.92 yr) is significantly older than the survivors (58.74 ± 11.94 yr) (p<0.001), while the LVEF is significantly smaller in the non-survivors. The decline of LVEF in the non-survivors group might be related to their older age. The analysis of parameters that can affect the mortality rate of the patients must take into account the effect of aging and declined LVEF on the all-cause mortality of the patients.

2. After a median follow up of 40.3 months, 37 (22.7%) patients died. A median follow up of 40 months, more than 3 years, seems to be too long to link between the abnormality in post-dialysis HRV parameters and the all-cause mortality of the patients. During the 40.3 months’ follow-up period, many factors might come in to affect the outcome of the patients. Would it be better to stratify the non-survivors according to their causes of death to find out the relation between the abnormality in their post-dialysis HRV parameters and the cause of death?

3. In the Abstract, the authors stated that post-dialysis HRV parameters including higher very low frequency (VLF), higher normalized low frequency (nLF) and higher LF/HF ratio were the independent predictors associated with lower risk for all-cause mortality. Higher post-dialysis normalized high frequency (nHF) increased risk of mortality. Among VLF, nLF, LF/HF and nHF, which one is the most important independent predictor of all-cause mortality? Why did the authors present the survival curve of post-dialysis LF/HF in Table 2 only, without the survival curves of VLF, nLF, and nHF?

Minor concerns

1. Line 107-108: The authors said: “Patients who were lost to follow-up were censored at the last visit date. Patients who underwent kidney transplantation were censored at the transplant date.” What is meant by “censored”?

2. Line 131-132: The authors stated: ”The use of beta-blockers was significantly higher in the non-survivors than in the survivors (82.86% vs 61.48%, p=0.025).” Does this means that the use of beta-blockers is detrimental to patients with ESRD?

6. PLOS authors have the option to publish the peer review history of their article (what does this mean?). If published, this will include your full peer review and any attached files.

Reviewer #1: **Yes: **Hideki ISHII

Reviewer #2: No

Reviewer #3: **Yes: **Cheng-Deng Kuo

---

## [Author Response · Author response to Decision Letter 0]

24 Jan 2023

Responses to Revision Letter 

Dear Academic Editor

Manuscript: PONE-D-22-29950

Title: Predictive Value of Heart Rate Variability on Long-Term Mortality in End-Stage Renal Disease on Hemodialysis

Thank you very much for your consideration to revise our manuscript. As very instructive comments from reviewers, we have response to their comments as following.

Reviewer #1: In the study, the authors evaluated the predictive value of HRV on all-cause mortality and also explored the proper timing of HRV assessment in patients on dialysis.

It is well known that autonomic disturbance is common in such population. The findings are interesting. However, the study enrolled a very small number of patients.

In addition, important information are lacking in the present form.

Answer: We would like to thank the reviewer for those constructive comments and suggestions. We have revised our manuscript according to those comments. Point-by-point responses to the comments are listed below.

Specific comments:

1. How did the authors check LVEF? If UCG was used, other important parameters such as LVMI, E/e’ LAD and so on might affect clinical outcomes in patients on dialysis.

Answer: We would like to thank the reviewer for giving us helpful comments on this issue. Before starting hemodialysis, a bedside echocardiogram was performed on the same day as Holter monitoring. Due to time constraints, we only measured the LVEF using a modified biplane Simpson's technique. There were no other echocardiographic parameters recorded. As a result, we lack information on diastolic function and left ventricular mass. Non-survivors had significantly lower LVEF, and it has been included in the multivariable analysis for mortality. (Table 3) (Page 12). The information on LVEF measurement was also added in the manuscript. Now it reads “Before starting hemodialysis, a bedside echocardiogram was performed. The modified biplane Simpson's technique was used to calculate the left ventricular ejection fraction (LVEF).” (Page 4 line 103-105).

2. QT duration is associated with incidence of critical arrhythmia.

Answer: Thank you the reviewer for this concern. QT interval can be measured in the 12-leads electrocardiogram (ECG). However, we did not perform the 12-leads ECG in our study due to the financial issue. This is one of major limitation of our study and we have added this issue in the limitation part of the main manuscript. It now reads “Some potential variables associated with cardiovascular mortality were not collected including the residual renal function and QT interval.” (Page 16 line 324-325)

3. Etiology of renal disease should be added.

Answer: We would like to thank the reviewer for this suggestion. We have added the etiology of ESKD in table 1 as reviewer suggested (Page 7). The most common etiology of ESKD in our study was diabetic nephropathy followed by hypertension.

4. How did the authors measure blood pressure and laboratory data? Please explain.

Answer: We would like to thank the reviewer for this useful comment. Blood pressure was measured before hemodialysis in a seated position after resting for more than 5 minutes using validated automatic oscillometric device (OMRON HEM-7121 automatic blood pressure monitor). Blood samples for complete blood count, creatinine, electrolytes, and albumin were collected before the initiation of hemodialysis. Blood pressure was repeated after completion of hemodialysis session in the same manner. Blood for creatinine and electrolytes were collected again after dialysis as part of hemodialysis care protocol. This information was added to the main manuscript. (Page 4 line 97-103)

5. During follow-up phase, death occurred in 37 patients. Please add precise causes of deaths in such patients.

Answer: We would like to thank the reviewer for this useful comment. Of the 37 patients who died, 7 were from cardiovascular causes, 9 were from infection, 3 were from metabolic causes, 13 had an unknown cause and 5 had other causes. For the cardiovascular causes, 3 had heart failure, 3 had sudden cardiac death and 1 had acute myocardial infarction. These data were added to the manuscript in Page 6 line 149-152.

6. Please add definitions of comorbidity.

Answer: We would like to thank the reviewer for this comment. Comorbidities are defined in our study as other health conditions that may affect the outcome of ESKD patients. Other atherosclerotic diseases and chronic lung disease, such as hypertension, diabetes, coronary artery disease, peripheral arterial disease, dyslipidemia, and COPD, are included, as shown in Table 1 (Page 7).

Reviewer #2: Thank you for the opportunity to review this study.

This study examines the predictive ability of heart rate variability before, during, and after dialysis for mortality, respectively.

Answer: We would like to thank the reviewer for those constructive comments and suggestions. We have revised our manuscript according to those comments. Point-by-point responses to the comments are listed below.

It is a very interesting study, but I have a few concerns.

1. After line 57, the authors introduce a previous study on the association between heart rate variability and mortality in hemodialysis patients. The authors state that the results are inconsistent, but it would be better to further organize the information from existing studies. Fig. 4 of the meta-analysis (reference #5) cited by the authors shows that studies with long-term measurements show less heterogeneity. The results of studies with short-term measurements similar to this study are inconsistent. If this study focuses its research question on which of the short-term measurements has better predictive ability, then it would be better to organize the information by focusing on timing rather than discussing parameter issues, so that the reader can clearly see what is known and what is not known. If the focus is on the question of whether a study has better predictive ability, then organizing the information to focus on timing rather than discussing parameter issues should clearly communicate to readers what is known and what is not known.

Answer: Thank you reviewer for this important concern. Our study measured HRV in a short-term period (10 minutes pre-dialysis, 4-hr during dialysis and 10 minutes after hemodialysis). The aims of our study were to find the predictive parameters for mortality and to determine the appropriate time points (before, during, and after hemodialysis) of HRV assessment. Therefore, we have added more details of previous studies focusing on time points of HRV assessment, as reviewer suggested. Now it reads “Additionally, the appropriate time point for HRV measurement was not well established. There were only two previous studies that performed repeated HRV measurement on the dialysis day. Chen et al. measured HRV before and 30 minutes after hemodialysis session. They found that the change of LF between the pre-hemodialysis and post-hemodialysis predicted overall and cardiovascular mortality [9]. While a more recent study by Chang et al performed the repeated HRV measurements before and during hemodialysis and found that HRV parameters during hemodialysis were predictive for cardiovascular mortality [10]. To the best of our knowledge, there was no previous study that performed repeated HRV measurement at 3-time points (pre-dialysis, during dialysis and post-dialysis). As a result, the aims of our study were to determine the best timing for HRV measurement by performing repeated HRV analysis at 3-time points and to determine the best predictive HRV parameter on all-cause mortality.” (Page 3 Line 79-85 to Page 4 line 86-89). The details of HRV timing were also included in the discussion part “There were only two previous studies that performed repeated HRV measurements on the dialysis day [9, 10]. The first study by Chen et al. calculated a difference in HRV parameters between pre-dialysis and post-dialysis, decreased in LF was shown to be predictive of overall and cardiovascular mortality [10]. Another study by Chang et al. performed the repeated HRV measurement at pre-dialysis and during dialysis but did not perform the post-dialysis measurement [9]. They identified the decreased VLF and LF/HF ratio during dialysis as the independent predictors for cardiovascular mortality. Our study was the first study that measured HRV parameters at 3 time points including pre-dialysis, during and post-dialysis.” (Page 15 Line 288-295). 

2.The authors chose to employ a variable with a p-value <0.10 in the univariate analysis as the adjustment variable in the multivariate analysis; Table 1 shows that diastolic blood pressure was not employed for post-dialysis, despite the fact that p<0.10 for both pre-dialysis and post-dialysis. Why were both not entered into the model? If these show multicolinearity, show the data. Also, it seems odd that they used pre-dialysis diastolic blood pressure to adjust for post-dialysis heart rate variability when the final results show that post-dialysis HRV has better predictive power.

Answer: We would like to thank the reviewer for raising this point. We did not add post-dialysis diastolic blood pressure (DBP) in the model because it had some collinearity with pre-dialysis DBP. According to the analysis by Pearson’s correlation, pre- and post-dialysis DBP had an R value of 0.50, indicating some correlation (Figure in response to reviewer file). However, we have re-analyzed by adding the post-dialysis DBP into the model as reviewer suggested and the result showed non-significant different from the prior model (Table in response to reviewer file). Therefore, we decided to use only pre-dialysis DBP as an adjustment variable because it represented the patients’ condition prior to dialysis.

Figure: Scatter plot between pre-dialysis DBP and post-dialysis DBP with R-value = 0.50 

3. The authors mention in the discussion that the amount of fluid removal by dialysis is related to heart rate volatility, but there are several studies that show that residual renal function and the amount of fluid removed are themselves related to mortality. Shouldn't these be included as adjustment variables?

Answer: We would like to thank the reviewer for this comment. We have analyzed the amount of fluid removal as reviewer suggested and found no difference between survivors and non-survivors (mean fluid removal per session in survivors = 2449.57 ml and non-survivors = 2294 ml; p = 0.38). These data have been added to table 1. (Page 7). As the amount of fluid removal was not different between the two groups, we did not include this factor as adjustment variable. It is the limitation of our study that we do not have the data of residual renal function. This has been added to the limitation part of the main manuscript. Now it reads “Some potential variables associated with cardiovascular mortality were not collected including the residual renal function and QT interval.” (Page 16 line 324-325).

4. The authors present the results of Table 3 and conclude that the post-dialysis parameters were predictive. The authors seem to have reached this conclusion because only the post-dialysis parameter showed a statistically significant difference, but it is questionable whether this is due to the size of the effect size or the sample size. Second, it is also questionable whether this conclusion can be reached without a direct statistical comparison of each HRV parameter. If the authors want to compare the predictive power of the parameters, they should obtain the c-index and compare each parameter to each other.

Answer: We would like to thank the reviewer for this useful suggestion. We have compared the predictive value of each HRV parameter using the area under receiver operating characteristic curve (AuROC) as reviewer suggested. The results were shown in table 4. (Page 14). The results were also described in the main manuscript. Now it reads “We compared the predictive value of HRV parameters by preforming AuROC. The parameters were directly compared at the same period (pre-dialysis, during dialysis or post-dialysis). Each parameter was also compared at 3 time points. The results were presented in table 4. There was no significant difference in the AuROC before and during dialysis. Interestingly, there was a significant difference in the AuROC at post-dialysis period. The AuROC of VLF for survival was highest compared to other HRV parameters at post-dialysis period (AuROC 0.71; 95% CI; 0.62-0.79; p<0.001). Post-dialysis VLF had highest AuROC for survival compared to pre-dialysis VLF and VLF during dialysis (p=0.008).” (Page 13 line 253-260). In addition, we have included these AuROC results in the Abstract section, which reads, “The area under receiver operating characteristic curve (AuROC) of VLF for survival was highest compared to other HRV parameters at post-dialysis period (AuROC 0.71; 95% CI; 0.62-0.79; p<0.001).” (Page 2 line 50-52).

5.There is a typo error in the reference 7.

Answer: This has been corrected in the revised manuscript as suggested by the reviewer.

Reviewer #3: This study evaluated the predictive value of heart rate variability (HRV) on all-cause mortality and explore the proper timing of HRV assessment. Clinical parameters and all-cause mortality were analyzed using cox proportional hazard regression. They found that post-dialysis HRV parameters including higher very low frequency (VLF), higher normalized low frequency (nLF) and higher LF/HF ratio were the independent predictors associated with lower risk for all-cause mortality, and that higher post-dialysis normalized high frequency (nHF) increased the risk of mortality. They concluded that HRV parameters predicted all-cause mortaliy in end-stage renal disease patients ESRD. This is an interesting study. However, some concerns need to be addressed.

Answer: We would like to thank the reviewer for those constructive comments and suggestions. We have revised our manuscript according to those comments. Point-by-point responses to the comments are listed below.

Major concerns

1. Table 1. The age of non-survivors (70.32 ± 13.92 yr) is significantly older than the survivors (58.74 ± 11.94 yr) (p<0.001), while the LVEF is significantly smaller in the non-survivors. The decline of LVEF in the non-survivors group might be related to their older age. The analysis of parameters that can affect the mortality rate of the patients must take into account the effect of aging and declined LVEF on the all-cause mortality of the patients.

Answer: We would like to thank the reviewer for this useful suggestion. Several factors can affect the mortality rate of patients. The non-survivors were significantly older and had lower LVEF compared to survivors. Therefore, both age and LVEF were included as adjustment variables along with other variables with p-value <0.10. The results of multivariable adjustment and the details of all variables included in multivariable analysis were shown in table 3. (Page 12) 

2. After a median follow up of 40.3 months, 37 (22.7%) patients died. A median follow up of 40 months, more than 3 years, seems to be too long to link between the abnormality in post-dialysis HRV parameters and the all-cause mortality of the patients. During the 40.3 months’ follow-up period, many factors might come in to affect the outcome of the patients. Would it be better to stratify the non-survivors according to their causes of death to find out the relation between the abnormality in their post-dialysis HRV parameters and the cause of death?

Answer: We would like to thank the reviewer for this useful comment. Of the 37 patients who died, 7 were from cardiovascular causes, 9 were from infection, 3 were from metabolic causes, 13 had an unknown causes and 5 had other causes. This information has been added to the main manuscript at Page 6 line 149-151. However, due to the small number of deaths in each category, there is an inadequate statistical power to determine a relationship between the cause of death and the HRV parameters.

3. In the Abstract, the authors stated that post-dialysis HRV parameters including higher very low frequency (VLF), higher normalized low frequency (nLF) and higher LF/HF ratio were the independent predictors associated with lower risk for all-cause mortality. Higher post-dialysis normalized high frequency (nHF) increased risk of mortality. Among VLF, nLF, LF/HF and nHF, which one is the most important independent predictor of all-cause mortality? Why did the authors present the survival curve of post-dialysis LF/HF in Table 2 only, without the survival curves of VLF, nLF, and nHF?

Answer: We would like to thank the reviewer for this useful comment. To answer which parameter is the most important predictor, we have further analyzed and compared the predictive value of each HRV parameter using the area under receiver operating characteristic curve (AuROC) and found that post-dialysis VLF was the best predictive parameter for survival. The results of AuROC of each HRV parameter were presented in table 4. (Page 14). The results were also described in the main manuscript. Now it reads “We compared the predictive value of HRV parameters by performing AuROC. The parameters were directly compared at the same period (pre-dialysis, during dialysis or post-dialysis). Each parameter was also compared at 3 time points. The results were presented in Table 4. There was no significant difference in the AuROC before and during dialysis. Interestingly, there was a significant difference in the AuROC at post-dialysis period. The AuROC of VLF for survival was highest compared to other HRV parameters at post-dialysis period (AuROC 0.71; 95% CI; 0.62-0.79; p<0.001). Post-dialysis VLF had highest AuROC for survival compared to pre-dialysis VLF and VLF during dialysis (p=0.008).” (Page 13 line 253-260). Also, we have added this finding in the conclusion of the abstract, which reads, “VLF measured at post-dialysis exhibited best predictive value for survival in chronic hemodialysis patients.” (Page 2 line 52-53).

We initially decided to only show the post-dialysis LF/HF ratio graph because it had the lowest hazard ratio for mortality. After analyzing the AuROC, we discovered that post-dialysis VLF had the highest predictive value for survival (AuROC = 0.71). As a result, as suggested by the reviewer, we created the graph of VLF and other post-dialysis parameters in Figure 1 (Page 11). Lower post-dialysis VLF, nLF, and LF/HF ratios were predictive of mortality (HR, 3.50; p<0.001, HR 2.25; p=0.02, and HR 2.55; p=0.006, respectively). nHF at the median or higher level was also predictive of mortality (HR 2.25; p=0.02). 

Minor concerns

1. Line 107-108: The authors said: “Patients who were lost to follow-up were censored at the last visit date. Patients who underwent kidney transplantation were censored at the transplant date.” What is meant by “censored”?

Answer: We would like to thank the reviewer for asking this question. The primary outcome of our study was performed by Cox’s proportional hazard regression which is a time-to-event analysis (time to death). Censoring occurs when an event is not observed for reasons such as study termination or the patient leaving the study prior to experiencing the event. The time to event was calculated from the enrolled date through the censored date. We followed all of the patients until the end of September 2021, when the study was terminated. If the patients lived until the end of September 2021, the time-to-event was calculated on that date (censored), and they were classified as no-event. If they were lost to follow up, the time to event was calculated using the last-visit date (censored), and they were classified as no-event. For patients who received a kidney transplant, the time to event was calculated through the transplant date (censored) and they were classified as no-event. Time to event was calculated using the death date for patients who died and were defined as having an event.

2. Line 131-132: The authors stated: “The use of beta-blockers was significantly higher in the non-survivors than in the survivors (82.86% vs 61.48%, p=0.025).” Does this means that the use of beta-blockers is detrimental to patients with ESRD?

Answer: We would like to thank the reviewer for asking this interesting point. As the reviewer mentioned, the non-survivor group used beta-blockers at a higher rate than the survivor group. As a result, beta-blocker use may be associated with an increase in mortality. However, several factors influence beta-blocker use in ESKD. Other diseases that may necessitate the use of beta-blockers include coronary artery disease and heart failure. To answer this question, potential factors associated with increased risk of death including beta-blocker use were included in the multivariable analysis model, as shown in Table 3. (Page 12)

---

## [Editor Report · Decision Letter 1]

14 Feb 2023

Predictive Value of Heart Rate Variability on Long-Term Mortality in End-Stage Kidney Disease on Hemodialysis

PONE-D-22-29950R1

Dear Dr. Noppakun,

We’re pleased to inform you that your manuscript has been judged scientifically suitable for publication and will be formally accepted for publication once it meets all outstanding technical requirements.

Kind regards,

Eyüp Serhat Çalık

Academic Editor

PLOS ONE
---

## [Editor Report · Acceptance letter]

16 Feb 2023

PONE-D-22-29950R1 

Predictive Value of Heart Rate Variability on Long-Term Mortality in End-Stage Kidney Disease on Hemodialysis 

Dear Dr. Noppakun:

I'm pleased to inform you that your manuscript has been deemed suitable for publication in PLOS ONE. Congratulations! Your manuscript is now with our production department. 

Kind regards, 

on behalf of

Dr. Eyüp Serhat Çalık 

Academic Editor

PLOS ONE